# Measurement Report: Seasonal trends and chemical speciation of chromium(III/VI) in different fractions of urban particulate matter – a case study of Radom, Poland

Monika Łożyńska[1], Marzena Trojanowska[2], Artur Molik[2], Ryszard Świetlik[2]

[1]Łukasiewicz Research Network – Institute for Sustainable Technologies, Bioeconomy and Ecoinnovation Centre, 26-600 Radom, Pulaskiego 6/10, Poland
[2]Casimir Pulaski Radom University, Faculty of Applied Chemistry, 26-600 Radom, Chrobrego 27, Poland

*Correspondence to*: Łożyńska Monika (monika.lozynska@itee.lukasiewicz.gov.pl)

**Abstract.** The paper assesses chromium occurrence in urban particulate matter: $PM_{10}$, $PM_{2.5}$, $PM_1$, and $PM_{0.25}$ over a calendar year. The seasonality of both pseudo-total chromium content and its valence speciation are studied. Airborne aerosol was sampled in the city of Radom (Poland) using a Hi-Flow MOUDI Model 130 cascade impactor from Copley. $Cr_{tot}$ concentrations in the PM fractions investigated ranged from 0.08 to 4.09 $ng/m^3$. The results point to a seasonality of $Cr_{tot}$ concentration changes in PM. The concentration of $Cr_{tot}$ in $PM_{10}$ was maximum in winter ($2.23\pm0.53$ $ng/m^3$ on average), and averaged $1.71\pm0.83$ $ng/m^3$ in the whole measurement period. The average Cr(VI) concentration did not exceed 0.40 $ng/m^3$ and was maximum in winter, too (max. 1.354 $ng/m^3$). The Cr(VI) share in PM in the particular seasons varied from a low in summer (9.1% of $Cr_{tot}$) to a high in winter (40% of $Cr_{tot}$). The carcinogenic risk for the urban residents based on the Cr(VI) concentration in $PM_{10}$ was below the US EPA threshold for significant risk (between $1\cdot10^{-6}$ and $1\cdot10^{-4}$) and amounted to between $1.11\cdot10^{-6}$ and $5.78\cdot10^{-6}$ for children and from $3.69\cdot10^{-6}$ to $1.92\cdot10^{-5}$ for adults. The non-carcinogenic health risk caused by the presence of $Cr_{tot}$ was also lower than the safe level of 1 - the HQ values for both adults and children ranged from $1.32\cdot10^{-2}$ to $3.92\cdot10^{-2}$. The studies presented in the manuscript fill the research gap of $Cr_{tot}$ and Cr(VI) measurements in particulate matter of different sizes in the air of a medium-sized city in central Poland.

## 1 Introduction

Air quality has continued to deteriorate lately, especially in urban and industrial areas. Particulate matter (PM) is a major pollutant whose substantial quantities are emitted into the air, affecting adversely both the climate and all parts of the environment (Fang et al., 2005; Wagner et al., 2008; Arhami et al., 2017). Particulate matter, considered one of the most dangerous environment pollutants (Park et al., 2009; Waheed et al., 2011; Izhar et al., 2016), has a complex and heterogeneous composition, dependent on the season, source of emissions, and weather conditions. Knowledge of the composition, concentration, and sources of particulate matter suspended in the air is of great importance to residents of urban-industrial areas, since breathing these particles can increase mortality or morbidity due to respiratory and pulmonary conditions. The

particulate matter's coarse fraction (2.5-10 µm) is assumed to be of natural origin, while the fine fraction (0.1–2.5 µm, especially the particles of less than 1 µm) mostly comes from anthropogenic processes (Nocoń et al., 2018). The distribution of particulate matter particle sizes and their loading with different chemical compounds determine the aerosol's effect on human health and the environment. Particulate matter with grain diameters below 2.5 µm is particularly dangerous to living
organisms as it penetrates alveoli pulmonis and thence potentially enters the bloodstream (Wagner et al., 2008; Pan et al., 2015). For this reason, this particulate matter fraction is the most common object of environmental studies (Feng at al., 2009; Rogula- Kozłowska et al., 2013a; Zajusz-Zubek and Mainka, 2015; Li et al., 2016; Sah et al., 2019; Xie et al., 2019; Anake et al., 2020; Wu et al., 2021; Wang et al., 2023;) (Table S1, Supplementary Material). Researchers frequently examine $PM_{10}$ too (e.g. Richter et al., 2007; Catrambone et al., 2013; Huang et al., 2014a; Pandey et al., 2017; Rubio et al., 2018; Conca et al.,
2020). The finer fractions, e.g. $PM_1$, are investigated far more seldom (Rogula-Kozłowska et al., 2013b; Zajusz-Zubek et al., 2015; Zajusz-Zubek, 2017). Scientists rarely assay two ($PM_{10}$ and $PM_{2.5}$: Canepari et al., 2009; Dos Santos et al., 2009; Jan et al., 2018; Gunchin et al., 2021) or more grain fractions ($PM_1$, $PM_{2.5}$, and $PM_{10}$: Samara and Voutsa, 2005; Rogula-Kozłowska et al., 2013b; Rogula-Kozłowska et al., 2015) in parallel.

Various organic and inorganic compounds are also transported together within the particulate matter, some of which are toxic.
Special attention is paid to heavy metals (Wagner et al., 2008; Pan et al., 2015; Samara et al., 2016) that may contribute to oxidative DNA damage and cause carcinogenic lesions in effect (Somers, 2011; IARC, 2012; Arhami et al., 2017). Chromium occurs in the air in two valence states: Cr(III) and Cr(VI), greatly varying in their physical and chemical properties and toxicity. Chromium(III) is a microelement necessary for living organisms, whereas chromium(VI) is toxic and classified as a carcinogen (Katz, 1991; Barceloux, 1999; Kotaś and Stasicka, 2000). Epidemiological research has shown a close connection between
chromium(VI) exposure and lung cancer (IARC, 2012). Chromium air presence has been studied a lot given its toxicity (Nriagu and Nieboer, 1988; Nusko and Heumann, 1997; Świetlik et al., 2011; Tirez et al., 2011; Torkmahalleh et al., 2013; Huang et al., 2014a, 2014b; Kang et al., 2016; Widziewicz et al., 2016; Molik et al., 2018; Nocoń et al., 2018). Regionally, natural sources account for 30-40% of total chromium emissions (Pacyna, 1986; Kotaś and Stasicka, 2000). Authors report that fuel combustion in stationary sources is the chief source of Cr emissions in Europe and more than a half of all anthropogenic
sources (Pacyna et al., 2007). In Poland, the burning of fuels is responsible for 60% of all the Cr emissions to the environment (Ministry of Climate and Environment, 2020).

Nriagu et al. (1988) report an average atmospheric concentration of Cr starting from 1 ng/m$^3$ in rural areas to 10 ng/m$^3$ in polluted urban areas. Świetlik and Trojanowska (2022) review the average Cr in the urban air of diverse regions globally. Given the commonly encountered pollution levels worldwide, $Cr_{tot}$ in the urban air averaged 13±2 ng/m$^3$, whereas $Cr_{tot}$ in
heavily polluted areas was many times higher, 350±70 ng/m$^3$ (Świetlik and Trojanowska, 2022). Average Cr(VI) concentration is far lower – 0.5 ng/m$^3$ (Seinfeld and Pandis, 2006). It is normally higher close to industrial sources (Proctor et al., 2021). According the US EPA National Air Toxics Assessment in 2017, the median, mean, and maximum Cr(VI) concentrations in the United States were 0.03 ng/m$^3$, 0.1 ng/m$^3$ and 3.18 ng/m$^3$, respectively (Proctor et al., 2021).

To assure legal protection against atmospheric chromium in Poland, reference values were set as follows: 20 μg/m$^3$ and 2.5 μg/m$^3$ for Cr(III) and Cr(VI) compounds and 4.6 μg/m$^3$ and 0.4 μg/m$^3$ for Cr(VI) over 1 h and 1 year, respectively (The Minister of the Environment Regulation, 2010).

Although chromium occurrence in urban air has been extensively studied and a range of publications have appeared recently (Catrambone et al., 2013; Widziewicz et al., 2016; Nocoń et al., 2018), investigations of total Cr occurrence and its valence speciation in particulate matter of different particle sizes are still limited. We have only examined Cr(III/VI) speciation in total suspended particulate (TSP) before (Świetlik et al., 2011). Our current study is designed to: (I) assess chromium occurrence and speciation in urban particulate matter fractions of varied particle sizes (PM$_{10}$, PM$_{2.5}$, PM$_1$, PM$_{0.25}$); (II) examine the fluctuations and seasonality of Cr concentrations over one year, and (III) estimate health risk caused by inhalation exposure to airborne Cr in two different exposure schemes: 1) the risk that comes from the Cr$_{tot}$ ambient concentrations; 2) the risk that comes exclusively from Cr(VI) species. Radom is an interesting location for such research as the atmospheric chromium levels are a result not only of an aged urban structure relying on private hard coal heating, considerable road transit, and the operation of multiple metal working factories, but also tanneries clustered in the region for more than 70 years.

These are the results of the first-ever hexavalent chromium (Cr(VI)) measurements in the particulate matter of different sizes in the air of a medium-sized city in central Poland.

## 2 Experimental

### 2.1 Sampling area and the collection of PM samples

The airborne particulate matter was sampled in Radom, a medium-sized town (a population of 196 000, an area of 112 km$^2$) in central Poland, 100 km south of Warsaw. The local sources of chromium emissions are: road traffic, coal burning in homes, coal-fired municipal heating plants, tanneries and multiple metalworks. The sampling points of the suspended airborne particulate matter were located on the University of Radom premises. The area is adjacent to a housing estate with a prevalence of private houses heated with traditional coal furnaces, but not under an immediate impact of local sources of industrial emissions or streets with heavy traffic.

The airborne particulate matter was sampled in spring (variable weather: cool, sunny, wet; the heating campaign until the middle of April), summer (sunny, warm weather), autumn (variable weather: sunny, warm or cool, wet; heating campaign from the middle of October), and winter (cold, overcast and wet weather; heating campaign). Averaged weather conditions for each weekly sampling cycle are presented in Table S2 (Supplementary Material). A Hi-Flow MOUDI Model 130 cascade impactor from Copley was used, positioned at a height of 1.5 m above the ground level. It enabled the separation of four grain fractions of atmospheric dust: PM$_{10}$, PM$_{2.5}$, PM$_1$ and PM$_{0.25}$. The grain fractions of atmospheric particulate matter were sampled from February 2020 to April 2021 in 29 weekly cycles. A total of 116 samples were collected. The sampling time averaged 70 h ± 4 h. The sampling rate was maintained at approximately 6 m$^3$/h. The airborne particulate matter was collected on cellulose filters (POCH). The filters were weighed before and after sampling, the precision was 0.01 mg (Microbalance MX5 Mettler

Toledo) in a temperature and relative humidity controlled environment (20±3°C and 50±10%, respectively). Then the filters were stored in a plastic CD-case. Before the analysis, the filters with the deposit were sectioned into four equal parts. The samples were only withdrawn when there was no precipitation.

## 2.2 The determination of the contents of the particular airborne particulate matter fractions

The concentrations of the specific particulate matter fractions were calculated by dividing the difference between filter weight prior to and following the exposure by the mean air flow at the time of atmospheric aerosol sampling.

## 2.3 The determination of pseudo-total chromium

In order to assay $Cr_{tot}$ by means of graphite furnace atomic absorption spectrometry (GF-AAS), the particulate matter samples collected on filters were mineralised using microwave energy. A quarter of the deposited filter was placed in a 100 mL Teflon

vessel with 5 mL 65% $HNO_3$ and 3 mL 30% $H_2O_2$. The samples were digested in a microwave oven (Milestone MLS 1200 Mega) according to the following program: 1) 6 min., 250 W; 2) 1 min., 0W; 3) 6 min., 400 W; 4) 6 min., 650 W; 5) 6 min., 250 W; 6) 5 min., ventilation. After the digestion, the solutions were filtered through a 0.45 μm Millex-HV syringe filter (Millipore), transferred them into a polypropylene volumetric flask, and diluted to 25.0 mL with deionized water.

A Perkin-Elmer 3100 AAS graphite furnace system equipped with HGA 600 and an AS-60 autosampler were used for the

determination of pseudo-total chromium concentration. The standard chromium solution (J.T. Baker Inc.) at a concentration of 1000 μg/mL served to calibrate the device. Limit of detection (LOD) (instrumental) was found to be 0.2 μg/L of Cr or 0.03 $ng/m^3$ expressed as the concentration of Cr in the air. The quality of the chromium results was characterized with a recovery test of the CRM BCR-701 (river sediment). It was found to be 259±18 mg/kg $Cr_{tot}$ (n = 3), while the certified value is 272±20 mg/kg, hence the recovery for extractable Cr was 95.2%.

## 2.4 The determination of chromium(VI)

The filters with the collected airborne particulate matter were extracted as per a modified US EPA 3060A standard by means of alkaline digestion (US EPA, 1996). The modification was necessary given the low chromium(VI) contents in the suspended particulate matter samples. For this purpose, a quarter of the exposed filter was placed in the Teflon vessel, 5 mL of 0.056 M $Na_2CO_3$/0.08 M NaOH was added and extracted in a microwave oven MLS 1200 Mega (3 min, 400W). The leachates were

then centrifuged (MPW 342) and filtered through a 0.45 μm Millex-HV syringe filter (Millipore) into 10 mL test tubes.

Cr(VI) in the leachates was determined using the technique of catalytic cathodic stripping voltammetry with the adsorption of Cr(III)-DTPA complexes (CCSV-DTPA) (Li and Xue, 2001). The voltammograms were recorded with a Trace Analyzer Model 394 connected to a hanging mercury drop working electrode Model 303A SMDE (EG&G Princeton Applied Research Electrochemical).

The supporting electrolyte (5 mL water, 500 μL saturated solution of $KNO_3$, 200 μL 0.1 M AcONa, 200 μL 0.1 M DTPA) and a 100 μL sample of Cr(VI) extract were transferred into an electrochemical cell, pH was adjusted to 6.00±0.05, and the solution

was purged for 4 min with high purity argon. Deposition on a mercury drop was carried out for 60s at -0.95 V. Then the stirring was discontinued and an equilibration time of 10 s was allowed. The differential pulse scan was carried out from -0.95 V to -1.65 V, the pulse height was 50 mV and the rate 6 mV/s. The peak corresponding to the reduction of adsorbed Cr(III)-DTPA complex appeared at -1.20 V to -1.30 V. The chromium(VI) concentration was assayed using the technique of double spike-standard addition (2-10 μL, depending on the expected analyte concentration) with a concentration of 0.2 mg/L Cr(VI). LOD (instrumental) was found to be 2.9 μg/L or 0.010 $ng/m^3$ expressed as the concentration of Cr(VI) in the air. CMR061-030 (sandy loam) was used as the certified reference material to validate the analytical procedure. It was found to equal 239.3 mg/kg Cr(VI) (n =3), the certified value was 241.00±9.00 mg/kg, hence the recovery of 99.3%. The recovery is high enough (Cr - 95.2% and Cr(VI) - 99.3%) and our results are not related to the enforcement analysis, which is why the data presented in this report are uncorrected for sample recovery efficiency.

## 3 Results and discussion

### 3.1 The mass concentration of PM

The concentrations of the following particulate matter fractions were assayed: $PM_{10}$, $PM_{2.5}$, $PM_1$, and $PM_{0.25}$. The World Health Organization (WHO) claims that there is no evidence of a safe exposure level or threshold below which there are no adverse effects of exposure to particulate matter (WHO, 2013).

In Radom, the $PM_{10}$ concentration ranged widely from 5.2 to 68.2 $μg/m^3$ (40±17 $μg/m^3$ on average). The $PM_{2.5}$ concentrations were lower, from 3.1 to 59.2 $μg/m^3$ (33±15 $μg/m^3$ on average). The concentrations of finer particulate matter were in the following ranges: $PM_1$ from 2.4 to 45.4 $μg/m^3$ (26±13 $μg/m^3$ on average), and $PM_{0.25}$ – from 1.3 to 27.6 $μg/m^3$ (12.6±6.6 $μg/m^3$ on average) (Table 1). Comparable concentrations of urban $PM_{2.5}$ are reported by other authors, e.g., Budapest (Hungary) 23 $μg/m^3$, Istanbul (Turkey) 40 $μg/m^3$ (Szigeti et al., 2013), Tehran (Iran) 33±11 $μg/m^3$ (Arhami et al., 2017), Katowice (Poland) 31 $μg/m^3$ (Rogula-Kozłowska et al., 2013a).

These concentrations were far lower than reported from our earlier study of TSP – from 36 to 282 $μg/m^3$, average 110 $μg/m^3$ (Świetlik et al., 2011).

**Table 1. The concentrations of PM in the urban atmosphere.**

|  | $PM_{10}$ [$μg/m^3$] | $PM_{2.5}$ [$μg/m^3$] | $PM_1$ [$μg/m^3$] | $PM_{0.25}$ [$μg/m^3$] |
|---|---|---|---|---|
| Spring | | | | |
| Min | 5.2 | 3.1 | 2.4 | 1.3 |
| Max | 62.7 | 48.6 | 36.6 | 20.6 |
| Mean | 25 | 19 | 15 | 7.5 |
| SD | 21 | 17 | 13 | 7.2 |
| Summer | | | | |
| Min | 22.4 | 16.9 | 12.7 | 6.6 |
| Max | 43.3 | 33.1 | 22.9 | 12.0 |

| | | | | |
|---|---|---|---|---|
| Mean | 28.6 | 22.1 | 16.3 | 8.3 |
| SD | 7.9 | 6.2 | 4.1 | 2.0 |
| **Autumn** | | | | |
| Min | 31.1 | 23.6 | 17.9 | 7.6 |
| Max | 68.2 | 59.2 | 45.1 | 27.6 |
| Mean | 53 | 44 | 34.3 | 15.9 |
| SD | 10 | 10 | 8.3 | 5.5 |
| **Winter** | | | | |
| Min | 23.1 | 19.0 | 15.4 | 7.0 |
| Max | 60.9 | 51.8 | 45.4 | 24.2 |
| Mean | 45 | 39 | 32 | 15.8 |
| SD | 13 | 12 | 10 | 6.0 |

The average $PM_{10}$ and $PM_{2.5}$ concentrations in autumn ($53\pm10$ µg/m$^3$; $44\pm10$ µg/m$^3$) and winter ($45\pm13$ µg/m$^3$; $39\pm12$ µg/m$^3$) were in excess of the national (40 µg/m$^3$ and 20 µg/m$^3$, respectively) and European (40 µg/m$^3$ and 25 µg/m$^3$, respectively) air quality standards (The Minister of the Environment Regulation 2012; Directive 2024/2881). A relatively high share of $PM_{2.5}$ respirable fraction relative to $PM_{10}$ was found in the measurement period, 81% on average. Zajusz-Zubek and Mainka (2015) report a similar share of $PM_{2.5}$ (88%), although the particulate matter concentrations measured by them were somewhat lower: $PM_{10} - 25.58$ µg/m$^3$; $PM_{2.5} - 22.93$ µg/m$^3$ ($PM_1$ concentration $- 18.44$ µg/m$^3$) (Zajusz-Zubek and Mainka, 2015). The literature states $PM_{2.5}$ in most European locations constitutes 50 to 70% of $PM_{10}$ (WHO, 2013).

Seasonality had a great influence on the concentrations of particulate matter suspended in urban air (Figure S1, Supplementary Material). During the winter season, the concentrations of each fraction were nearly double those in summer. In the autumn, however, slightly higher concentrations of $PM_{10}$, $PM_{2.5}$ and $PM_1$ were recorded compared to the winter season (on average by 12%). Only $PM_{0.25}$ concentrations were steady in autumn and winter and stood at $15.9\pm5.5$ µg/m$^3$ and $15.8\pm6.0$ µg/m$^3$, respectively (Table 1).

The high airborne particulate matter concentrations in the urban area investigated during the winter season suggest municipal emissions including coal burning in household furnaces were the chief source of particulate matter pollution.

The higher suspended particulate matter concentrations in the winter season have also been found by others, e.g., Rogula-Kozłowska et al. (2013a) reports $PM_{2.5}$ concentrations in Zabrze and Katowice (Poland) usually ranging between 10 and 20 µg/m$^3$ and far greater in winter, from 50 to 110 µg/m$^3$ (Rogula-Kozłowska et al., 2013a). The higher suspended particulate matter concentrations in winter (pointing to a greater share of fuel combustion sources) have been noted by other authors, too (Krzemińska-Flowers et al., 2006; Widziewicz et al., 2016).

It should be pointed out that particulate matter pollution has been substantially reduced in Poland in recent years, owing to the application of state-of-the-art, efficient, and environment-friendly technological solutions. The continuing modernization of the energy, heating, and industrial sectors - such as the EU Clean Air Program (since 2018) and provincial anti-smog resolutions (since 2017) - along with improved fuel quality regulations (established by the Minister of Industry and the Minister of Climate and Environment regarding the quality of solid fuels since 2018), has led to a consistent reduction in the amount of

particulate matter pollution emitted into the air each year (EU Clean Air Program, 2024; Regulation of the Minister of Industry and the Minister of Climate and Environment, 2024).

### 3.2 Chromium concentration

Given the environmental aspect of this study, the environmentally available chromium content $Cr_{tot}$ (pseudo-total concentration
of Cr) in the PM samples was assayed. All the results for $Cr_{tot}$ concentrations in $PM_{10}$, $PM_{2.5}$, $PM_1$ and $PM_{0.25}$ are listed in Table S3 (Supplementary Material).

$Cr_{tot}$ concentrations in the particulate matter fractions studied ranged from 0.08 to 4.09 $ng/m^3$. Like in the case of particulate matter concentrations, the results were several times lower than the chromium concentrations given in our previous work for the non-industrial zone (15 $ng/m^3$ on average), when we studied TSP (Świetlik et al., 2011). In Poland, the concentrations of
185 Cr in the PM, similar to the particulate matter concentrations, decrease every year thanks to the use of state-of-the-art, efficient and environmentally friendly technological solutions.

The results indicate a marked impact of the winter season on $Cr_{tot}$ concentrations in all the PM fractions (Figure 1).

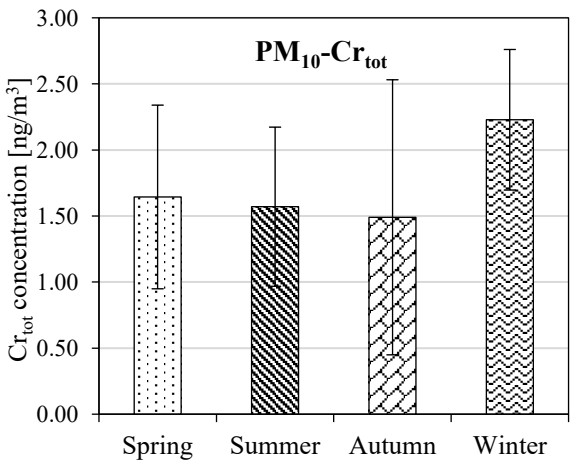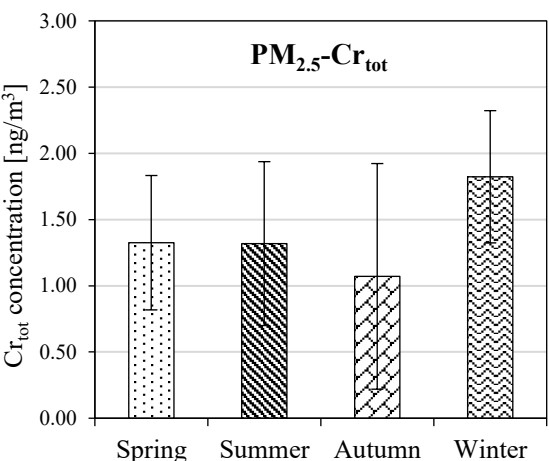

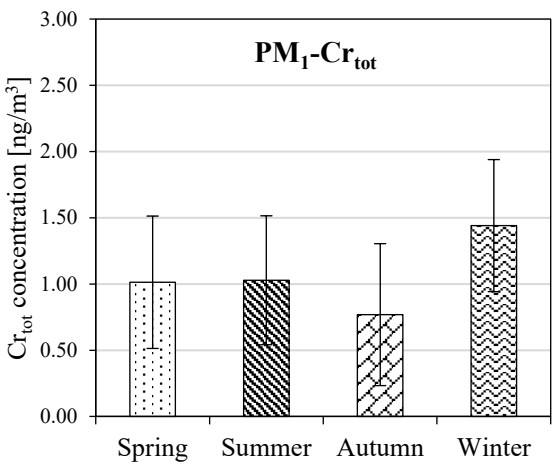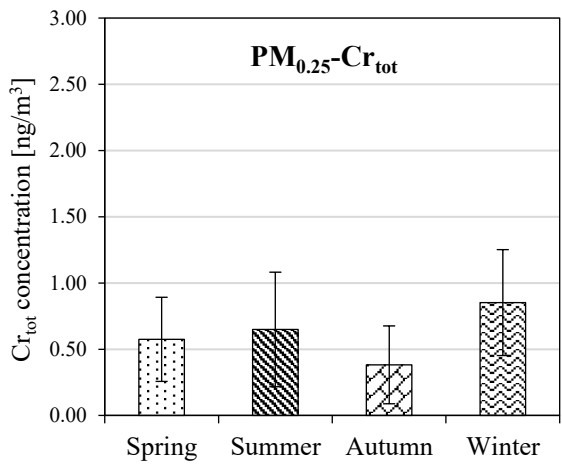

**Figure 1. The concentrations of pseudo-total chromium in $PM_{10}$, $PM_{2.5}$, $PM_1$ and $PM_{0.25}$ depending on the seasons.**

The maximum $Cr_{tot}$ concentration relating to $PM_{10}$ was found in winter ($2.23\pm0.53$ ng/m$^3$ on average), whereas it averaged $1.71\pm0.83$ ng/m$^3$ in the entire measurement period. The mean $Cr_{tot}$ concentrations relating to $PM_{10}$ in Radom were similar to those determined in other cities in Europe in urban areas: Edinburgh (U.K.) 1.6 ng/m$^3$ (Heal et al., 2005), Katowice (Poland) 2.39 ng/m$^3$ (Rogula-Kozłowska, 2015), Rome (Italy) 2-5 ng/m$^3$ (Catrambone et al., 2013).

The $PM_{2.5}$ fraction contains approximately 80% of the total chromium content. A similar correlation was observed for the PM concentration in the winter season, where 86% of $PM_{10}$ was the $PM_{2.5}$ fraction (Table 1). The $PM_{2.5}$ fraction is assumed to be the result of emissions from anthropogenic sources (Nocoń et al., 2018). The sampling point was located near a busy street and a single-family housing estate (from the west and south-west) and far away from an industrial zone (approx. 8-10 km from the south and south-west). Considering that Radom is a city with prevailing westerly winds, especially in autumn and winter (WeatherSpark, 2025), it can be supposed that municipal emissions, mainly stationary coal combustion sources, and road traffic are probably the main factors influencing the measured chromium concentrations.

$Cr_{tot}$ concentrations in $PM_{2.5}$, $PM_1$ and $PM_{0.25}$ were: $1.38\pm0.69$ ng/m$^3$, $1.06\pm0.55$ ng/m$^3$, $0.61\pm0.39$ ng/m$^3$, respectively (Table S3). The mean chromium concentrations relating to $PM_{2.5}$ in Radom were similar to those determined in other cities in Poland and globally: Zabrze $1.7\pm1.9$ ng/m$^3$ (Rogula-Kozłowska et al. 2013a); Warsaw $1.2\pm1.4$ ng/m$^3$ (Majewski and Rogula-Kozłowska, 2016); Wrocław $1.6\pm0.8$ ng/m$^3$ (Zwoździak et al., 2013); Łódź $2.82\pm0.34$ ng/m$^3$ ($PM_3$, Krzemińska-Flowers et al., 2006); Budapest (Hungary) 1.4 ng/m$^3$ (Muránszky et al., 2011), Istanbul (Turkey) 2.8 ng/m$^3$ (Szigeti et al., 2013); Rome (Italy) 3.72 ng/m$^3$ (Canepari et al., 2009). The concentrations are far greater in industrial areas, e.g., Guangzhou (China) 7.693 ng/m$^3$ (Feng et al., 2009); Ewekoro (Nigeria) 11.4 ng/m$^3$ (Anake et al., 2020); Agra (India) 19.3 ng/m$^3$ (Sah et al., 2019); Nanjing (China) 26.61 ng/m$^3$ (Li et al., 2016).

In spring and summer, $Cr_{tot}$ concentrations for the particulate matter fractions were similar and averaged: $PM_{10}$ – $1.64\pm0.70$ and $1.57\pm0.60$ ng/m$^3$, $PM_{2.5}$ – $1.33\pm0.51$ and $1.32\pm0.62$ ng/m$^3$, $PM_1$ – $1.01\pm0.50$ and $1.03\pm0.49$ ng/m$^3$, $PM_{0.25}$ – $0.57\pm0.32$

ng/m$^3$ and 0.65±0.43 ng/m$^3$, respectively (Table S3). They were maximum in winter: PM$_{10}$ 2.23±0.53 ng/m$^3$; PM$_{2.5}$ 1.82±0.50 ng/m$^3$; PM$_1$ 1.44±0.50 ng/m$^3$; PM$_{0.25}$ 0.85±0.40 ng/m$^3$ (Figure 1). The higher Cr$_{tot}$ concentrations in winter are corroborated by other authors (Krzemińska-Flowers et al., 2006; Zwoździak et al., 2013; Widziewicz et al., 2016).

Based on the differences of Cr$_{tot}$ concentrations in the particular fractions (PM$_{10}$ and PM$_{2.5}$; PM$_{2.5}$ and PM$_1$; PM$_1$ and PM$_{0.25}$),
its concentrations in the fractions were calculated as follows: PM$_{2.5-10}$, PM$_{1-2.5}$, PM$_{0.25-1}$ and PM$_{0.25}$ (Table S4, Supplementary Material). Regardless of the season, Cr$_{tot}$ content was always highest in the finest fraction, PM$_{0.25}$ (Figure S2, Supplementary Material).

### 3.3 The speciation of chromium

Chromium speciation was also assayed in all the fractions of airborne particulate matter. Cr(VI), as a particularly harmful
metal, is classified by the International Agency for Research on Cancer (IARC) as a Group 1 carcinogen (IARC, 2023). Authors report most airborne Cr(VI) (60–70%) comes from anthropogenic sources (e.g. metallurgy, refractory production or chemical processing industries) (Kang et al., 2016; Wang et. al., 2020). Emissions associated with solid fuel combustion are of great significance as well (Nriagu and Pacyna, 1988). Combustion processes, mostly of anthropogenic origin, or gas to particle conversions of atmospheric substances produce small aerosol particles, whereas larger particles are formed by mechanical
processes, such as wind erosion of soil or dust emission from public traffic (Nusko and Heumann, 1997). Cr(VI) presence in Radom's airborne aerosol may be chiefly a result of municipal (fuel burning in heating plants and household furnaces) and road traffic emissions. Industrial emissions are of lesser importance as the industrial sources are located far away from the sampling points (about 8-10 km).

Cr(VI) concentrations in the urban air of Radom were quite varied and ranged from undetectable levels (<LOD) to 1.354 ng/m$^3$
(Table S3). Average Cr(VI) concentrations did not exceed 0.400 ng/m$^3$ in the entire measurement period, though, and equalled: PM$_{10}$ – 0.38 ng/m$^3$; PM$_{2.5}$ – 0.32 ng/m$^3$, PM$_1$ – 0.26 ng/m$^3$ and PM$_{0.25}$ – 0.16 ng/m$^3$, respectively. Similar Cr(VI) concentrations in airborne particulate matter are reported by other authors: Wilmington (USA) 0.5-1.0 ng/m$^3$ (PM$_{2.5}$) (Khlystov and Ma, 2006), New Jersey (USA) 0.86-1.56 ng/m$^3$ (PM$_{10}$) (Huang et al., 2014b), Beijing (China) 0.006–0.266 ng/m$^3$ (PM$_{2.5}$) (Wang et. al., 2023), although higher concentrations are also found, e.g., the Flemish region (Belgium) 1.2–5.2 ng/m$^3$ (PM$_{10}$) (Tirez
et al., 2011).

Our investigation has shown 78-90% of total Cr(VI) content (depending on the season) was found in PM$_{2.5}$ (83% on average). Like in the case of Cr$_{tot}$, a distinct effect of the heating in winter season on Cr(VI) concentrations in PM could be noted. This was true of all the grain fractions (Figure 2).

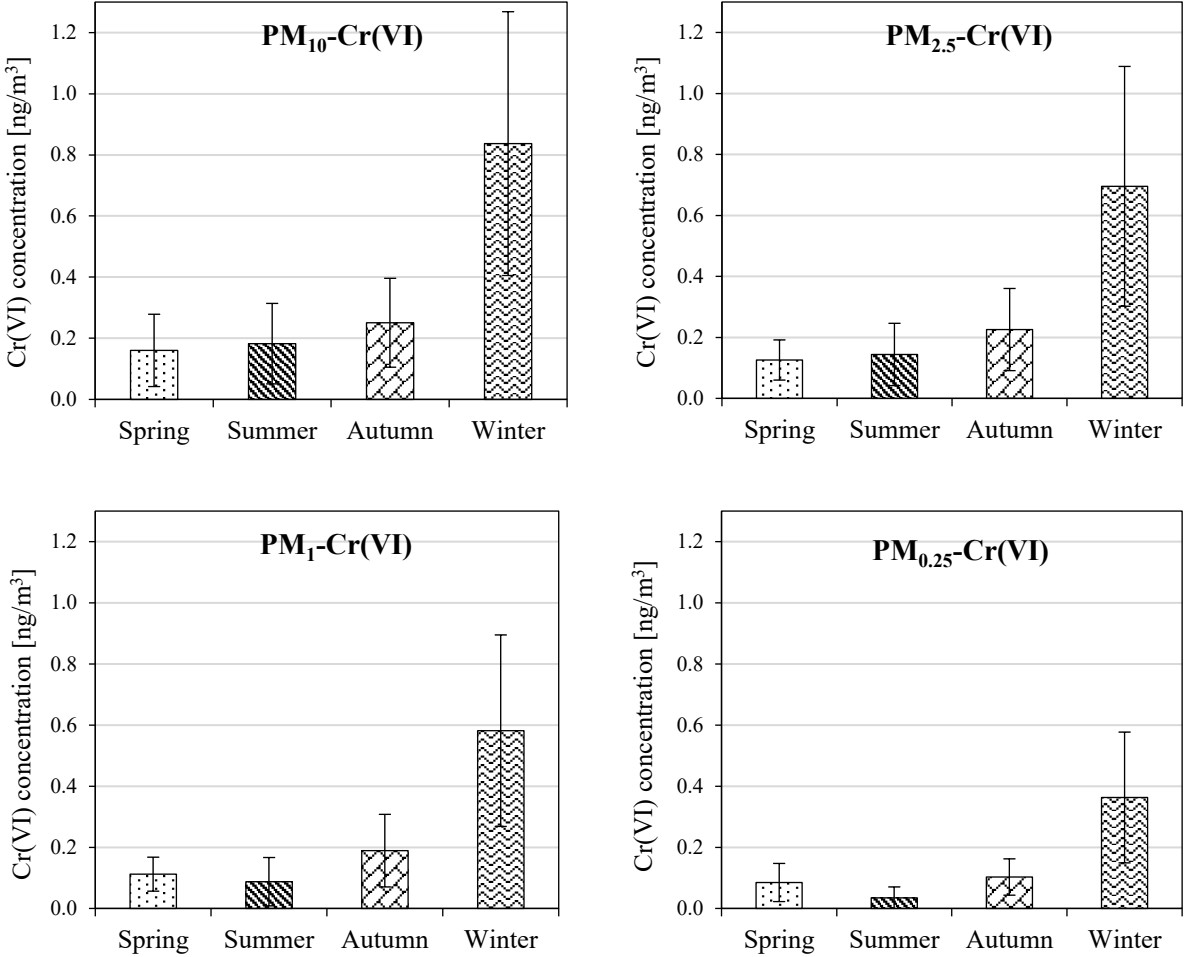

**Figure 2. The concentrations of Cr(VI) in PM₁₀, PM₂.₅, PM₁ and PM₀.₂₅ depending on the seasons.**

Cr(VI) concentrations were maximum in winter (the heating campaign) and averaged: $PM_{10}$ – 0.84±0.43 ng/m³, $PM_{2.5}$ – 0.70±0.39 ng/m³, $PM_{1}$ – 0.58±0.31 ng/m³, $PM_{0.25}$ – 0.36±0.21 ng/m³. Maximum Cr(VI) concentrations in $PM_{10}$ reached 1.354 ng/m³. For the sake of legal protection against airborne chromium(VI), a reference value of 0.4 μg/m³ (for 1 year) was introduced in Poland (The Minister of the Environment Regulation, 2010). The World Health Organization (WHO) recommends the baseline limit of hexavalent chromium with an excess lifetime risk (RR values) of 1:10,000, 1:100,000 and 1:1,000,000 to be 2.5, 0.25, 0.025 ng/m³, respectively (WHO, 2000).

Regardless of the particulate matter fraction, chromium(VI) accounted for approximately 20% of $Cr_{tot}$ content in the particulate matter ($PM_{10}$ - 19.3%, $PM_{2.5}$ – 19.9%, $PM_{1}$ – 21.1%, $PM_{0.25}$ – 22.4%) during the whole measurement period (Table 2). Cr(VI) share in the particular seasons varied, however. It was minimum in summer, averaging 9.1%, somewhat greater in spring (11.3%), to rise to 22.7% (from 18.1% to 26.9%) in the autumn. It was maximum in winter, constituting 40% of total chromium

on average (from 37.6% to 42.6%). Our results are not different from those reported in the literature, e.g., Bell and Hipfner (1997) and Talebi (2003) claim that circa 20% airborne chromium was Cr(VI), Nusko and Heumann (1997) give approximately 30%. Tirez et al. (2011) reported that, in the case of $PM_{2.5}$ from coal combustion, the share of Cr(VI) ranged from 6 to 43% of the total Cr (Tirez et al., 2011). Torkmahalleh et al. (2013) state the average Cr(VI) to total Cr ratios varied from 1 to 30%, whereas Kang et al. (2016) say the concentration of Cr(VI) measured accounted for 0.7 to 9.4 percent of the total chromium level, which is a low percentage compared to those in other urban areas around the world.

**Table 2. The percentage share of chromium speciation.**

| Season | $PM_{10}$ | | $PM_{2.5}$ | | $PM_1$ | | $PM_{0.25}$ | |
|---|---|---|---|---|---|---|---|---|
| | Share of Cr(VI) [%] | Share of Cr(III) [%] | Share of Cr(VI) [%] | Share of Cr(III) [%] | Share of Cr(VI) [%] | Share of Cr(III) [%] | Share of Cr(VI) [%] | Share of Cr(III) [%] |
| Spring | 9.8 | 90.2 | 9.5 | 90.5 | 11.1 | 88.9 | 14.8 | 85.2 |
| Summer | 11.6 | 88.4 | 10.9 | 89.1 | 8.5 | 91.5 | 5.3 | 94.7 |
| Autumn | 18.1 | 81.9 | 21.1 | 78.9 | 24.6 | 75.4 | 26.9 | 73.1 |
| Winter | 37.6 | 62.4 | 38.1 | 61.9 | 40.4 | 59.6 | 42.6 | 57.4 |
| **Average value** | 19.3 | 80.7 | 19.9 | 80.1 | 21.1 | 78.9 | 22.4 | 77.6 |

Cr(III) averaged 79.3% of $Cr_{tot}$ (Table 2). It was maximum in summer (90.9% on average) and a little lower in spring (88.7%). It was minimum in winter, 60.3% of the total chromium content (Figure S3, Supplementary Material).

Based on the difference of Cr(VI) concentrations assayed in $PM_{10}$, $PM_{2.5}$, $PM_1$ and $PM_{0.25}$, its concentrations in: $PM_{2.5-10}$, $PM_{1-2.5}$, $PM_{0.25-1}$ and $PM_{0.25}$ were calculated (Table S4, Supplementary Material). Like in the case of $Cr_{tot}$, regardless of the season, Cr(VI) concentrations were normally maximum in the finest fraction, $PM_{0.25}$ (Figure S4, Supplementary Material). Some authors report that fine particulate matter, which can be carried over long distances, contain more Cr(VI) (Wang et al., 2023). Attempts at correlating $Cr_{tot}$ concentrations in $PM_{10}$ with its values in finer fractions: $PM_{2.5}$, $PM_1$, and $PM_{0.25}$, brought interesting results. The coefficients of determination ($R^2$) are relatively high: 0.9512 (for $PM_{10}$-$PM_{2.5}$), 0.8678 (for $PM_{10}$-$PM_1$), and 0.6283 (for $PM_{10}$-$PM_{0.25}$), mean $R^2 = 0.8158$ (Figure S5, Supplementary Material).

The correlation was even better for valence speciation. A clear dependence could be observed between the total Cr(VI) content in $PM_{10}$ and Cr(VI) content in finer fractions: $PM_{2.5}$, $PM_1$, and $PM_{0.25}$ (Figure S6, Supplementary Material). In this event, the coefficients of determination were greater than for $Cr_{tot}$: $R^2=0.9852$ (for $PM_{10}$-$PM_{2.5}$), 0.9715 (for $PM_{10}$-$PM_1$), and 0.9273 (and $PM_{10}$-$PM_{0.25}$), the average $R^2$ was 0.9613.

### 3.4 The health risk of inhalation exposure to chromium

The health risk of respiratory exposure to chromium was estimated in the airborne particulate matter investigated. The model of risk assessment recommended by the U.S. Environmental Protection Agency (US EPA, 2009) was applied. The risk for

both adults and children was estimated. The exposure concentration (*EC*) [µg/m³] of non-carcinogenic and carcinogenic chromium was calculated according to Eq. (1):

$$EC = \frac{C_{Cr} \cdot ET \cdot EF \cdot ED}{AT_n} \qquad (1)$$

where $C_{Cr}$ is the average concentration of $Cr_{tot}$ or Cr(VI) in atmospheric particulate matter [µg/m³]; *ET* - exposure time [hours/day], *EF* - exposure frequency [days/year], *ED* - exposure duration [years], and $AT_n$ is the averaging time of exposure [hours].

As recommended by the US EPA 2009 (Part F), a residential scenario could consist of inhalation exposure for up to 24 hours per day, up to 350 days per year for 6 to 30 years. The residential exposure parameters used in this study are listed in Table S5 (Supplementary material).

The non-carcinogenic health risk for chromium was evaluated by means of hazard quotient (*HQ*) using Eq. (2):

$$HQ = \frac{EC}{RfC_{Cr} \cdot 1000} \qquad (2)$$

where $RfC_{Cr}$ is the inhalation reference concentration for Cr [mg/m³]. Cancer risk (*CR*) was calculated using Eq. (3):

$$CR = EC \cdot IUR_{Cr} \qquad (3)$$

where $IUR_{Cr}$ is the inhalation unit risk for Cr(VI) [(µg/m³)$^{-1}$]. The values of $RfC_{Cr}$ and $IUR_{Cr}$ were cited from the Regional Screening Levels (RSL) Tables for US EPA Region 9 (US EPA 2022).

The acceptable carcinogenic risk ranges from $1 \cdot 10^{-6}$ (1 per 1 000 000) to $1 \cdot 10^{-4}$ (1 per 10 000) (US EPA, 1989). A carcinogenic risk value above the upper limit ($1 \cdot 10^{-4}$) suggests that chromium(VI) in atmospheric particulate matter poses a substantial risk of carcinogenic effects in the future from lifetime exposure, while values below the lower limit ($1 \cdot 10^{-6}$) do not pose a significant risk. *HQ* below one suggests no significant risk of non-carcinogenic effects. If *HQ* is equal to or more than 1, the non-carcinogenic effects are possible in the future.

The estimated potential non-carcinogenic (HQ) and carcinogenic (CR) risk of inhalatory exposure to $Cr_{tot}$ and Cr(VI) present in PM$_{10}$ for urban residents is shown in Table 3. Both were maximum in winter, when $Cr_{tot}$ concentrations become highest. In the light of the standard interpretation, however, regardless of the season, the carcinogenic risk to residents based on Cr(VI) concentration in PM$_{10}$ was below the US EPA threshold for significant risk (between $1 \cdot 10^{-6}$ and $1 \cdot 10^{-4}$) and amounted to between $1.11 \cdot 10^{-6}$ and $5.78 \cdot 10^{-6}$ for children and from $3.69 \cdot 10^{-6}$ to $1.92 \cdot 10^{-5}$ for adults. Even in the case of maximum Cr(VI) concentration in PM$_{10}$ during the winter season, the estimated carcinogenic risk to the population of Radom was lower than the upper limit: CR=$9.24 \cdot 10^{-6}$ for children and CR=$3.12 \cdot 10^{-5}$ for adults. However, although the risk of cancer in children is generally low, it can be misleading because exposures during development can have disproportionately dramatic effects in adulthood.

**Table 3. Carcinogenic and non-carcinogenic risk from chromium via inhalation exposure to the airborne particles of PM$_{10}$.**

| | Carcinogenic risk - CR | Non-carcinogenic risk - HQ |
|---|---|---|

| Season | Cr(VI) $[ng/m^3]$ | Children | Adult | $Cr_{tot}$ $[ng/m^3]$ | Children | Adult |
|---|---|---|---|---|---|---|
| Average concentration of chromium in $PM_{10}$ in each season | | | | | | |
| Spring | 0.1604 | $1.11 \cdot 10^{-6}$ | $3.69 \cdot 10^{-6}$ | 1.64 | $1.57 \cdot 10^{-2}$ | $1.57 \cdot 10^{-2}$ |
| Summer | 0.1823 | $1.26 \cdot 10^{-6}$ | $4.19 \cdot 10^{-6}$ | 1.57 | $1.51 \cdot 10^{-2}$ | $1.51 \cdot 10^{-2}$ |
| Autumn | 0.2508 | $1.73 \cdot 10^{-6}$ | $5.77 \cdot 10^{-6}$ | 1.38 | $1.32 \cdot 10^{-2}$ | $1.32 \cdot 10^{-2}$ |
| Winter | 0.8369 | $5.78 \cdot 10^{-6}$ | $1.92 \cdot 10^{-5}$ | 2.23 | $2.14 \cdot 10^{-2}$ | $2.14 \cdot 10^{-2}$ |
| Av. | | $2.47 \cdot 10^{-6}$ | $8.21 \cdot 10^{-6}$ | | $1.64 \cdot 10^{-2}$ | $1.64 \cdot 10^{-2}$ |
| Maximum chromium concentration in $PM_{10}$ throughout the measurement period | | | | | | |
| | 1.3544 | $9.24 \cdot 10^{-6}$ | $3.12 \cdot 10^{-5}$ | 4.09 | $3.92 \cdot 10^{-2}$ | $3.92 \cdot 10^{-2}$ |

The estimated non-carcinogenic inhalation risk from chromium for the residents of Radom posed little risk, either. The HQ values calculated based on the total Cr concentration in $PM_{10}$ were lower than the safe level (HQ = 1) and ranged from $1.32 \cdot 10^{-2}$ to $2.14 \cdot 10^{-2}$, indicating little non-carcinogenic risk from chromium. The non-carcinogenic health risk based on the maximum $Cr_{tot}$ concentration in $PM_{10}$ was low, too, as it was not more than one (Table 3). These risk values are no different from those found in the literature, e.g., HQ in Italy was in the range $1.5 \cdot 10^{-2} – 5.4 \cdot 10^{-2}$ (Diana et al., 2023).

EU air quality directives do not provide for separate normative values for chromium. Directive 2024/2881 specifies quality objectives for arsenic, cadmium, lead, mercury, nickel, and PAHs, but does not cover chromium (Directive 2024/2881). Within the framework of EU policy, chromium(VI) is regulated mainly in the context of occupational safety – the binding occupational exposure limit has been set at $0.005 mg/m^3$ by 2025 (Directive 2017/2398). For comparison, the highest obtained concentration of Cr(VI) in $PM_{10}$ ($1.3544 ng/m^3$) is almost three orders of magnitude lower than this value. Consequently, the estimated health risk of chromium is low and does not raise significant health concerns.

In conclusion, the estimated potential non-carcinogenic (HQ) and carcinogenic (CR) risk of inhalatory exposure to Cr and Cr(VI), respectively, present in $PM_{10}$ for the urban residents of Radom indicates that risks of adverse health effects from chromium and chromium(VI) exposure in Radom are relatively low and complies with the applicable WHO and EU regulatory framework (WHO, 2021; Directive 2017/2398).

## 4 Conclusion

The mean concentrations of $PM_{10}$ ($40 \pm 17 \mu g/m^3$) and $PM_{2.5}$ ($33 \pm 15 \mu g/m^3$) in the city were above the national ($40 \mu g/m^3$ and $20 \mu g/m^3$, respectively) and European ($40 \mu g/m^3$ and $25 \mu g/m^3$, respectively). The total chromium content in $PM_{10}$ ranged widely from 0.56 to $4.09 ng/m^3$ and averaged $1.71 \pm 0.83 ng/m^3$. The mean Cr(VI) concentration in $PM_{10}$ was $0.38 ng/m^3$ in the entire measurement period. Chromium(VI) accounted for circa 20% of $Cr_{tot}$ content in the particulate matter. The seasonality of $Cr_{tot}$ and Cr(VI) concentration changes could be noted. The concentrations were maximum in the winter (heating) season:

2.23 ng/m$^3$ and 0.84 ng/m$^3$, respectively, most likely a result of a greater share of the air polluted by fuel combustion sources in the air emissions of this type of pollution. The share of Cr(VI) in PM in the individual seasons varied as well: it was minimum in summer (9.1% of Cr$_{tot}$) and maximum in winter (40% of Cr$_{tot}$).

Out of the airborne particulate matter fractions investigated (PM$_{2.5-10}$, PM$_{1-2.5}$, PM$_{0.25-1}$, PM$_{0.25}$), both Cr$_{tot}$ and Cr(VI) concentrations were maximum in the finest fraction – PM$_{0.25}$. Toxicity related to transition metals is greater for fine than for coarse dust since, once they penetrate the respiratory system, the finest particles reach pulmonary alveoli, where 60-80% of a transmitted element enters the blood (Rogula-Kozłowska, 2013b). The health risk to urban residents, both carcinogenic and non-carcinogenic, is estimated to be highest in winter as Cr$_{tot}$ and Cr(VI) concentrations reach their top values. Regardless of the season, however, these risk levels to the residents of Radom were within the acceptable risk range.

Research into the forms of atmospheric Cr is limited and atmospheric Cr(VI) needs to be further investigated.

**Data availability**

Data sets are available at https://doi.org/10.5281/zenodo.14808852 (Łożyńska, M., et al., 2025)

**CRediT authorship contribution statement**

**M.Ł.** - Data curation, Formal analysis, Methodology, Writing – original draft, Writing – review and editing, Conceptualization, Investigation, Validation; **M.T.** – Data curation, Methodology, Writing – original draft, Writing – review and editing, Conceptualization, Investigation, Validation; **A.M.** - Data curation, Methodology, Visualization, Investigation; **R.Ś.** – Conceptualization, Methodology, Project administration, Supervision.

**Declaration of Competing Interest**

The authors declare that they have no conflict of interest.

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
