# Peer review of "Measurement Report: Seasonal trends and chemical speciation of chromium(III/VI) in different fractions of urban particulate matter – a case study of Radom, Poland"

_EGUsphere, 2025_

## Author Response (AR1)

**Response on RC1**

about manuscript:

"Measurement Report: Seasonal trends and chemical speciation of chromium(III/VI) in different fractions of urban particulate matter – a case study of Radom, Poland" Monika Łożyńska, Marzena Trojanowska, Artur Molik, Ryszard Świetlik.

Thank you very much for the suggestions and comments contained in the review. All suggested changes have been included in the manuscript. Answers to the comments are presented below:

**Specific Comments:**

**Abstract**

- Line 15: The language "The concentration was maximum in winter" is vague. Please specify which measurement ($PM_{10}$? $Cr_{tot}$?) you are referring to.

Authors' response:

As recommended by the reviewer, the sentence has been corrected. Current version (lines 15-16): The concentration $Cr_{tot}$ in $PM_{10}$ was maximum in winter ($2.23\pm0.53$ ng/m$^3$ on average), and averaged $1.71\pm0.83$ ng/m$^3$ in the whole measurement period.

- Line 18: "varied a lot" needs to be replaced with specific numbers. I suggest combining with the following sentence that contains details.

Authors' response:

As suggested by the reviewer, the sentences have been corrected. Current version (lines 17-18): The Cr(VI) share in PM in the particular seasons varied a lot, minimum in summer (9.1% of $Cr_{tot}$) and a maximum in winter (40% of $Cr_{tot}$).

- Lines 19 – 21: Consider removing the term "acceptable risk" and replacing its use with specific estimates of risk or HQ.

Authors' response:

The authors agree with the reviewer's suggestion. The sentence has been reworded. Current version (lines 18-22): The carcinogenic risk for the urban residents based on the Cr(VI) concentration in $PM_{10}$ was below the US EPA threshold for significant risk (between $1\cdot10^{-6}$ and $1\cdot10^{-4}$) and amounted to between $1.11\cdot10^{-6}$ and $5.78\cdot10^{-6}$ for children and from $3.69\cdot10^{-6}$ to $1.92\cdot10^{-5}$ for adults. The non-carcinogenic health risk caused by the presence of $Cr_{tot}$ was also lower than the safe level of 1 - the HQ values for both adults and children ranged from $1.32\cdot10^{-2}$ to $3.92\cdot10^{-2}$.

**Introduction**

- Line 29: The term "agglomeration" does not have a clear meaning here. Change to "regional" or similar.

Authors' response:

At the reviewer's suggestion, the sentence has been corrected. Current version (lines 28-31): Knowledge of the composition, concentration, and sources of particulate matter suspended in the air is of great importance to residents of urban-industrial areas, since breathing these particles can increase mortality or morbidity due to respiratory and pulmonary conditions.

- Line 44: This sentence implies the toxic compounds are not necessarily within the particulate matter. Consider changing to clarify that PM is comprised of organic and inorganic compounds, some of which are toxic.

Authors' response:

As suggested by the reviewer, the sentence has been corrected. Current version (line 44): Various organic and inorganic compounds are also transported together with the particulate matter, some of which are toxic.

- Line 48: Change "harmfulness" to "toxicity" for clarification.

Authors' response:

At the reviewer's suggestion, the sentence has been corrected. Current version (line 50): Chromium air presence has been studied a lot, given its toxicity.

- Lines 50 – 52: This is the introduction of chromium to readers. Move these sentences to appear before the sentence starting with "Epidemiological" on line 46. Some rewording may be useful to aid in flow after making this change.

Authors' response:

At the reviewer's suggestion, the paragraph has been reworded. Current version (lines 44-53): Various organic and inorganic compounds are also transported together with the particulate matter, some of which are toxic. Special attention is paid to heavy metals (Wagner et al., 2008; Pan et al., 2015; Samara et al., 2016) that may contribute to oxidative DNA damage and cause carcinogenic lesions in effect (Somers, 2011; IARC, 2012; Arhami et al., 2017). Chromium occurs in the air in two valence states: Cr(III) and Cr(VI), greatly varying in their physical and chemical properties and toxicity. Chromium(III) is a microelement necessary for living organisms, whereas chromium(VI) is toxic and classified as a carcinogen (Katz, 1991; Barceloux, 1999; Kotaś and Stasicka, 2000). Epidemiological research has shown a close connection between chromium(VI) exposure and lung cancer (IARC, 2012). Chromium air presence has been studied a lot given its toxicity (Nriagu and Nieboer, 1988; Nusko and Heumann, 1997; Świetlik et al., 2011; Tirez et al., 2011; Torkmahalleh et al., 2013; Huang et al., 2014a, 2014b; Kang et al., 2016; Widziewicz et al., 2016; Molik et al., 2018; Nocoń et al., 2018).

- Line 57: Provide a reference to justify the claim of the natural range of chromium in ambient PM.

Authors' response:

At the reviewer's suggestion, a reference has been added. Current version (line 57): The natural air content of $Cr_{tot}$ is estimated to range from 0.1 ng/m$^3$ to 1 ng/m$^3$ (Nriagu et al., 1988).

- Line 74: For both instances, change "the risk comes" to "the risk that comes".

Authors' response:

As suggested by the reviewer, the sentence has been corrected. Current version (lines 71-75): Our current study is designed to: (I) assess chromium occurrence and speciation in urban particulate matter fractions of varied particle sizes (PM$_{10}$, PM$_{2.5}$, PM$_1$, PM$_{0.25}$); (II) examine the fluctuations and seasonality of Cr concentrations over one year, and (III) estimate health risk caused by inhalation exposure to airborne Cr in two different exposure schemes: 1) the risk that comes from the $Cr_{tot}$ ambient concentrations; 2) the risk that comes exclusively from Cr(VI) species.

- Lines 75 – 76: Consider changing "chromium environment pollution is" to "atmospheric chromium levels are".

Authors' response:

As recommended by the reviewer, the sentence has been corrected. Current version (lines 75-77): Radom is an interesting location for such research, as the atmospheric chromium levels are a result not only of an aged urban structure relying on private hard coal heating, considerable road transit, and the operation of multiple metal working factories, but also tanneries clustered in the region for more than 70 years.

**Experimental**

- Line 88: Please add a sentence after this one that clearly defines the "heating campaign" and the range of dates you consider to be under within this subset of your measurements. When reading through the text, it appears that this is synonymous with the winter season? If so, I would recommend only referring to the winter season instead of the heating campaign.

Authors' response:

'Winter season' and 'heating campaign' were treated as synonyms by the authors. The authors agree with the reviewer's suggestion. With this in mind, the entire manuscript has been edited (lines 159, 163, 165, 234).

- Lines 88 – 90: Please provide summary statistics from the nearest available weather station of temperature and humidity. Precipitation and wind data would be useful as well.

Authors' response:

Thank you for your comment. Averaged weather conditions for each weekly sampling cycle are presented in Table S2 (Supplementary Material), which has been added to the article. All changes resulting from the addition of the table have been made throughout the article.

- Line 94: Provide the standard deviation of sampling times after reporting the average sample time.

Authors' response:

As suggested by the reviewer, the standard deviation of sampling times has been added. Current version (lines 94-95): The sampling time averaged 70 h ± 4 h.

- Line 100: Change "particular" to "specific".

Authors' response:

As suggested by the reviewer, the sentence has been corrected. Current version (lines 101-102): The concentrations of the specific particulate matter fractions were calculated by dividing the difference between filter weight prior to and following the exposure by the mean air flow at the time of atmospheric aerosol sampling.

- Line 103: This is the first instance of the term "GF-AAS", write out the entire name of the analytical method.

Authors' response:

As the reviewer suggested, the abbreviation "GF-AAS" was developed. Current version (lines 104-105): In order to assay $Cr_{tot}$ by means of graphite furnace atomic absorption spectrometry (GF-AAS), the particulate matter samples collected on filters were mineralised using microwave energy.

- Line 111: Write out the entire term for "LOD" on first use.

Authors' response:

At the reviewer's suggestion, the LOD acronym has been expanded. Current version (lines 112-113): Limit of detection (LOD) (instrumental) was found to be 0.2 µg/L of Cr or 0.03 $ng/m^3$ expressed as the concentration of Cr in the air.

- Line 114: You mention the Cr recovery rate. Did you use this number to correct the results from your analysis to infer measured Cr loadings on your samples?

And

- Line 134: The dash used before the recovery percentage could imply a negative recovery to readers. Change the dash to "of" for clarity. Also, did you use this number to correct the results from your analysis to infer measured Cr(VI) loadings on your samples?

Authors' response:

Thank you for your comment. The hyphen has of course been replaced by "of". Regarding "recovery value", the use of recovery information in analytical measurements is often optional. In our case, we were guided by the suitability of the measurement data for the specific purpose. We decided to use the original data in the manuscript because the recovery is high enough (Cr - 95.2% and Cr(VI) - 99.3%) and our results are not related to the enforcement analysis (the difference between applying or not applying a correction factor to the measurement data does not mean that a legal limit is exceeded or that a result is in compliance with the limit).

**Results**

- Line 140: Change "fraction" to "concentration" as this is referring to the total $PM_{10}$ values.

Authors' response:

As suggested by the reviewer, the sentence has been corrected. Current version (line 141): In Radom, the $PM_{10}$ concentration ranged widely from 5.2 to 68.2 $\mu g/m^3$ ($40\pm17$ $\mu g/m^3$ on average).

- Line 151: EU limit for annual $PM_{2.5}$ should be 20 ug/$m^3$.

And

- Line 291: Same comment as Line 151: EU limit for annual $PM_{2.5}$ should be 20 ug/$m^3$.

Authors' response:

Thank you for your comment. The currently applicable annual limit for $PM_{2.5}$ concentration in the European Union is 25 $\mu g/m^3$. This value was established by Directive 2008/50. According to the new EU Directive 2024/2881, adopted in December 2024, the limit value has been maintained at the same level as the value to be achieved by 11 December 2026. However, from 2030, the limit value will be 10 $\mu g/m^3$.

Under Directive 2008/50, the 20 $\mu g/m^3$ value for $PM_{2.5}$ was considered as Stage 2 of the limit for the Average Exposure Indicator (AEI), with a planned attainment date of January 1, 2020. However, this stage was indicative and subject to a review by the European Commission, considering new information on health and environmental effects, technical feasibility, and Member States' experience in implementing air quality objectives. In practice, the 20 $\mu g/m^3$ value has not been formally implemented as a binding legal limit in EU Member States.

- Line 159: The parenthetical "(by an average of 12%)" is not clear what numbers you are comparing. Is this comparing the average decline of measured PM between autumn and winter? This may get clarified by defining the dates of the "heating campaign" as mentioned above.

Authors' response:

As recommended by the reviewer, the sentence has been edited. Current version (lines 159-162): During the winter season, the concentrations of each fraction were nearly double those in summer. In the autumn, however, slightly greater concentrations of $PM_{10}$, $PM_{2.5}$ and $PM_1$ were recorded compared to the winter season (on average by 12%). Only $PM_{0.25}$ concentrations were steady in autumn and winter and stood at $15.9\pm5.5$ $\mu g/m^3$ and $15.8\pm6.0$ $\mu g/m^3$, respectively (Table 1).

- Line 172: Consider changing "quite widely" to a more specific "by 50x".

Authors' response:

Thank you for your comment. The authors decided to remove the phrase 'quite widely'. Current version (line 174): $Cr_{tot}$ concentrations in the particulate matter fractions studied ranged from 0.08 to 4.09 ng/m$^3$.

- Lines 174 – 178: Are there any references you can provide on Poland's efforts in the last 15 years towards reducing PM or chromium concentrations? New emissions standards or fuel standards would be helpful to the reader to understand the substantial decrease in ambient chromium levels.

Authors' response:

Thank you for your comment. Appropriate references have been added. Current version (lines 178-183): The continuing modernization of the energy, heating, and industrial sectors - such as the EU Clean Air Program (since 2018) and provincial anti-smog resolutions (since 2017) - along with improved fuel quality regulations (established by the Minister of Industry and the Minister of Climate and Environment regarding the quality of solid fuels since 2018), has led to a consistent reduction in the amount of particulate matter pollution emitted into the air each year [EU Clean Air Program, 2024; Regulation of the Minister of Industry and the Minister of Climate and Environment, 2024].

- Line 184: The parenthetical "(like in the case of particulate matter)" is not clear. Please report the mass fraction of $PM_{10}$ found in $PM_{2.5}$ here, and refer readers to Table 1.

And

- Lines 184 – 185: This is too broad a claim to be supported by the data presented here. Change the phrasing to "are likely major influences on the measured chromium concentrations" or remove entirely.

Authors' response:

The authors agree with the reviewer's suggestions. The sentences have been improved according to the comments of both reviewers. Current version (lines 187-197): The maximum $Cr_{tot}$ concentration relating to $PM_{10}$ was found in winter (2.23±0.53 ng/m$^3$ on average), whereas it averaged 1.71±0.83 ng/m$^3$ in the entire measurement period. The mean concentrations relating to $PM_{10}$ in Radom were similar to those determined in other cities in Europe in urban areas: Edinburgh (U.K.) 1.6 ng/m$^3$ (Heal et al., 2005), Katowice (Poland) 2.39 ng/m$^3$ (Rogula-Kozłowska, 2015), Rome (Italy) 2-5 ng/m$^3$ (Catrambone et al., 2013).
The $PM_{2.5}$ fraction contains approximately 80% of the total chromium content. A similar correlation was observed for the PM concentration in the winter season, where 86% of $PM_{10}$ was the $PM_{2.5}$ fraction (Table 1). The $PM_{2.5}$ fraction is assumed to be the result of emissions from anthropogenic sources (Nocoń et al., 2018). The sampling point was located near a busy street and a single-family housing estate (from the west and south-west) and far away from an industrial zone (approx. 8-10 km from the south and south-west). Considering that Radom is a city with prevailing westerly winds, especially in autumn and winter (WeatherSpark, 2025), it can be supposed that municipal emissions, mainly stationary coal combustion sources, and road traffic are probably the main factors influencing the measured chromium concentrations.

Line 186: Total Cr concentrations were lower than what? If $PM_{10}$, $PM_{2.5}$ concentrations should always be lower than $PM_{10}$ as they are a subset.

Authors' response:

As suggested by the reviewer, the sentence has been corrected. Current version (lines 198-199): $Cr_{tot}$ concentrations in $PM_{2.5}$, $PM_1$ and $PM_{0.25}$ were: 1.38±0.69 ng/m$^3$, 1.06±0.55 ng/m$^3$, 0.61±0.39 ng/m$^3$, respectively (Table S2).

- Line 188: Delete "during the whole measurement time" as no references were provided for studies overlapping with your reported sample dates.

Authors' response:

As suggested by the reviewer, the phrase "during the whole measurement time" has been removed. Current version (lines 199-203): The mean chromium concentrations relating to $PM_{2.5}$ in Radom were similar to those determined in other cities in Poland and globally: Zabrze 1.7±1.9 ng/m$^3$ (Rogula-Kozłowska et al. 2013a); Warsaw 1.2±1.4 ng/m$^3$ (Majewski and Rogula-Kozłowska, 2016); Wrocław 1.6±0.8 ng/m$^3$ (Zwoździak et al., 2013); Łódź 2.82±0.34 ng/m$^3$ ($PM_3$, Krzemińska-Flowers et al., 2006); Budapest (Hungary) 1.4 ng/m$^3$, Istanbul (Turkey) 2.8 ng/m$^3$ (Szigeti et al., 2013); Rome (Italy) 3.72 ng/m$^3$ (Canepari et al., 2009).

- Line 205: Consider changing "carcinogenic factor" to "carcinogen".

Authors' response:

As recommended by the reviewer, the sentence has been edited. Current version (lines 216-217): Chromium speciation was also assayed in all the fractions of airborne particulate matter. Cr(VI), as a particularly harmful metal, is classified by the International Agency for Research on Cancer (IARC) as a Group 1 carcinogen (IARC, 2023).

- Line 212: It would be helpful to provide an order of magnitude of the distance between the sampling site and the industrial sources mentioned.

Authors' response:

As suggested by the reviewer, the distance between the sampling site and the industrial sources has been added. Current version (lines 222-225): Cr(VI) presence in Radom's airborne aerosol may be chiefly a result of municipal (fuel burning in heating plants and household furnaces) and road traffic emissions. Industrial emissions are of lesser importance as the industrial sources are located far away from the sampling points (about 8-10 km).

- Line 218: The Wang et al. reference specifies which PM size fraction is reported. It would be helpful to add these to the other references presented in this sentence.

Authors' response:

As recommended by the reviewer, the size fractions of PM have been added to the remaining literature references. Current version (lines 228-232): Similar Cr(VI) concentrations in airborne particulate matter are reported by other authors: Wilmington (USA) 0.5-1.0 ng/m$^3$ ($PM_{2.5}$) (Khlystov and Ma, 2006), New Jersey (USA) 0.86-1.56 ng/m$^3$ ($PM_{10}$) (Huang et al., 2014b), Beijing (China) 0.006–0.266 ng/m$^3$ ($PM_{2.5}$) (Wang et. al., 2023), although higher concentrations are also found, e.g., the Flemish region (Belgium) 1.2–5.2 ng/m$^3$ ($PM_{10}$) (Tirez et al., 2011).

- Line 220: Consider changing "was $PM_{2.5}$" to "was found in $PM_{2.5}$".

Authors' response:

As suggested by the reviewer, the sentence has been corrected. Current version (line 233): Our investigation has shown 78-90% of total Cr(VI) content (depending on the season) was found in $PM_{2.5}$ (83% on average).

- Line 223 (Figure 2): The y-axis values have too many significant digits. Remove the trailing 3 zeroes from each tick label for clarity (e.g., change "1.2000" to "1.2").

Authors' response:

As suggested by the reviewer, the number of significant numbers on the Y axis has been reduced.

- Lines 271 – 272: The statement that a risk above 1 per 10,000 means the atmospheric Cr(VI) is "very likely to develop cancer" is misleading. Use similar phrasing to the language in the next sentence on HQ:

A risk above 1 per 10,000 poses significant risk while risk below 1 per 10,000 does not pose significant risk.

Authors' response:

In accordance with the reviewer's comment, the sentences have been corrected. Current version (lines 283-287): The acceptable carcinogenic risk ranges from $1.10^{-6}$ (1 in 1,000,000) to $1.10^{-4}$ (1 in 10,000) (US EPA, 1989). A carcinogenic risk value above the upper limit ($1.10^{-4}$) suggests that chromium(VI) in atmospheric particulate matter is likely to cause carcinogenic effects in the future from lifetime exposure, while values below the lower limit ($1.10^{-6}$) do not pose a significant risk. HQ of less than one suggests no significant risk of non-carcinogenic effects. If the HQ is equal to or greater than 1, non-carcinogenic effects are possible in the future.

- Line 275: Change "Cr" to "Cr(VI)".

Authors' response:

As suggested by the reviewer, the sentence has been corrected. Current version (lines 288-289): The estimated potential non-carcinogenic (HQ) and carcinogenic (CR) risk of inhalatory exposure to Cr and Cr(VI) present in PM10 for urban residents is shown in Table 3.

- Lines 278 – 281: I would again caution the use of the term "acceptable" and suggest you simply compare to the stated values from the US EPA (i.e., carcinogenic risk from particle-bound Cr(VI) was below the US EPA threshold for significant risk.).
  and
- Line 279: Consider adding a sentence that states while cancer risks for children were generally low, they can be misleading as exposure during development can lead to outsized impacts in adulthood.

Authors' response:

As suggested by the reviewer, the sentences have been corrected. Current version (lines 288-297): The estimated potential non-carcinogenic (HQ) and carcinogenic (CR) risk of inhalatory exposure to Cr and Cr(VI) present in PM10 for urban residents is shown in Table 3. Both were maximum in winter, when Cr concentrations become highest. The World Health Organization (WHO) recommends the baseline limit of hexavalent chromium with an excess lifetime risk (RR values) of 1:10,000, 1:100,000, and 1:1,000,000 to be 2.5, 0.25, and 0.025 $ng/m^3$, respectively [WHO, 2000]. In the light of the standard interpretation, however, regardless of the season, the carcinogenic risk to residents based on Cr(VI) concentration in $PM_{10}$ was below the US EPA threshold for significant risk (between $1 \cdot 10^{-6}$ and $1 \cdot 10^{-4}$) and amounted to between $1.11 \cdot 10^{-6}$ and $5.78 \cdot 10^{-6}$ for children and from $3.69 \cdot 10^{-6}$ to $1.92 \cdot 10^{-5}$ for adults. Even in the case of maximum Cr(VI) concentration in $PM_{10}$ during the winter season, the estimated carcinogenic risk to the population of Radom was lower than the upper limit: CR=$9.24 \cdot 10^{-6}$ for children and CR=$3.12 \cdot 10^{-5}$ for adults. However, although the risk of cancer in children is generally low, it can be misleading because exposures during development can have disproportionately dramatic effects in adulthood.

- Line 284: Change the phrase "posed no threat" to "posed little risk" to reflect risks were low but not zero.

Authors' response:

As recommended by the reviewer, the sentence has been edited. Current version (line 300): The estimated non-carcinogenic inhalation risks from chromium for the residents of Radom posed little risk, either.

- Line 286: Same comment as line 284: change "no non-carcinogenic risk" to "little non-carcinogenic risk".

Authors' response:

As recommended by the reviewer, the sentence has been corrected. Current version (lines 300-302): The HQ values calculated based on the total Cr concentration in $PM_{10}$ were lower than the safe level (HQ = 1) and ranged from $1.32 \cdot 10^{-2}$ to $2.14 \cdot 10^{-2}$, indicating little non-carcinogenic risk from chromium.

**Conclusion**

- Line 302: The phrase "gravest" does not match the sentiment that it posed little to no risk. Change to "highest".

Authors' response:

As recommended by the reviewer, the sentence has been corrected. Current version (lines 327-328): The health risk to urban residents, both carcinogenic and non-carcinogenic, is estimated to be highest in winter as $Cr_{tot}$ and Cr(VI) concentrations reach their top values.

**Technical corrections:**

Thank you so much for all your suggestions. The authors agree with all the reviewer's comments in the "Technical Corrections" section. All suggested changes are incorporated into the manuscript. The entire article was also reviewed for the writing of the particulate matter sizes as subscripts ($PM_{10}$, $PM_{2.5}$, $PM_1$, $PM_{0.25}$).

*With gratitude*

*Monika Łożyńska*

Measurement Report: Seasonal trends and chemical speciation of chromium(III/VI) in different fractions of urban particulate matter – a case study of Radom, Poland
Monika Łożyńska
Marzena Trojanowska
Artur Molik
Ryszard Świetlik

**Response on RC2**

about manuscript:

"Measurement Report: Seasonal trends and chemical speciation of chromium(III/VI) in different fractions of urban particulate matter – a case study of Radom, Poland" Monika Łożyńska, Marzena Trojanowska, Artur Molik, Ryszard Świetlik.

Thank you very much for the valuable suggestions and comments contained in the review. All suggested changes have been included in the manuscript. Answers to the comments are presented below:

**Abstract**

- Line 15: Replace vague phrasing "The concentration was maximum in winter" with a quantified statement.

Authors' response:

As recommended by the reviewer, the sentence has been corrected. Current version (lines 15-16): The concentration $Cr_{tot}$ in $PM_{10}$ was maximum in winter ($2.23\pm0.53$ ng/m$^3$ on average), and averaged $1.71\pm0.83$ ng/m$^3$ in the whole measurement period.

- Lines 18–19: Combine and specify.

Authors' response:

As suggested by the reviewer, the sentences have been corrected. Current version (lines 17-18): The Cr(VI) share in PM in the particular seasons varied a lot, minimum in summer (9.1% of $Cr_{tot}$) and a maximum in winter (40% of $Cr_{tot}$).

- Lines 19–21: Replace "acceptable risk" with: "The hazard quotient (HQ) for Cr(VI) reached Z in winter, indicating potential non-carcinogenic risk (HQ > 1)."

Authors' response:

The authors agree with the reviewer's suggestion. The sentence has been reworded. Current version (lines 18-22): The carcinogenic risk for the urban residents based on the Cr(VI) concentration in $PM_{10}$ was below the US EPA threshold for significant risk (between $1\cdot10^{-6}$ and $1\cdot10^{-4}$) and amounted to between $1.11\cdot10^{-6}$ and $5.78\cdot10^{-6}$ for children and from $3.69\cdot10^{-6}$ to $1.92\cdot10^{-5}$ for adults. The non-carcinogenic health risk caused by the presence of $Cr_{tot}$ was also lower than the safe level of 1 - the HQ values for both adults and children ranged from $1.32\cdot10^{-2}$ to $3.92\cdot10^{-2}$.

**Introduction**

- Line 29: Clarify "agglomeration" eg.using"urban-industrial areas"to avoid ambiguity.

Authors' response:

At the reviewer's suggestion, the sentence has been corrected. Current version (lines 28-30): Knowledge of the composition, concentration, and sources of particulate matter suspended in the air is of great importance to residents of urban-industrial areas, since breathing these particles can increase mortality or morbidity due to respiratory and pulmonary conditions.

- Line 44: Revise to explicitly link PM and toxicity.

Authors' response:

As suggested by the reviewer, the sentence has been corrected. Current version (line 44): Various organic and inorganic compounds are also transported together with the particulate matter, some of which are toxic.

- Line 57: Add a reference for natural Cr levels.

Authors' response:

At the reviewer's suggestion, a reference has been added. Current version (line 57): The natural air content of $Cr_{tot}$ is estimated to range from 0.1 ng/m$^3$ to 1 ng/m$^3$ (Nriagu et al., 1988).

**Experimental Section**

- Line 88: Define "heating campaign" explicitly.

Authors' response:

Thank you for your comment. The authors used "winter season" and "heating campaign" incorrectly as synonyms. As suggested by Reviewer 1, the entire manuscript has been corrected to remove the phrase 'heating campaign' and replaced with 'winter season' (lines 159, 163, 165, 234).

- Line 94: The sampling time reported to be averaged 70h, then, does it mean that the sampling duration time is different in different samplers? More explanation should be provided.

And

- Line 94, The air rate was also not consistent during the whole experiment (in the range of 0.35-0.5 m3/min), then, how the four particle sizes (PM10,PM2.5,PM1 and PM0.5) were divided by the cascade impactor?

Authors' response:

Thank you for your comments. Appropriate corrections were made to the final version of the manuscript.

The sampling time in the measurement cycle varied slightly and averaged 70±4 h. This was due to the tendency of the pump used to overheat, which resulted in the need to switch it off periodically.

The flow ranges were mistakenly written incorrectly in the manuscript, for which we sincerely apologize. The air flow was 6 m$^3$/h. With the increase in the mass of PM particles deposited on the final filter (PM$_{0.25}$), the flow rate decreased and was adjusted throughout the measurement cycle to ensure a constant flow rate of 6 m$^3$/h.

However, the authors would like to add that the calculations of chromium and chromium(VI) concentrations were based on the results of GFAAS and CCSV analyses, the actual sampling time, and the flow rate of 6 m$^3$/h. Similarly, the calculation of PM content was based on the mass of each PM fraction, the actual sampling time, and the flow rate of 6 m$^3$/h. The results of these calculations are presented in Table 1 and Table S3 (Supplementary Material).

- Line 103: Define"GF-AAS". Address sampling biases (e.g., single-site data) and analytical constraints (e.g., GF-AAS detection limits for low-concentration samples).

Authors' response:

As the reviewer suggested, the abbreviation "GF-AAS" was developed. Current version (lines 104-105): In order to assay $Cr_{tot}$ by means of graphite furnace atomic absorption spectrometry (GF-AAS), the particulate matter samples collected on filters were mineralised using microwave energy.
Unfortunately, it was not possible to perform Cr analyses in several repetitions. This was because the filter was divided into 4 equal parts according to the construction of the cascade impactor nozzle, which consists of four quadrants. For this reason, the authors used the reference material to verify the correctness of the analytical methodology used to obtain reliable chromium concentrations. The high recovery values presented in the article confirmed the correctness of the applied methodology.

Considering that PM samples were collected in weekly cycles, the volume of air passed through was large and therefore the mass of collected PM was also large. As a consequence, the obtained chromium concentrations in solutions after mineralization were above the detection limit.

- Line 111: Define "LOD"

Authors' response:

At the reviewer's suggestion, the LOD acronym has been expanded. Current version (lines 112-113): Limit of detection (LOD) (instrumental) was found to be 0.2 µg/L of Cr or 0.03 ng/m$^3$ expressed as the concentration of Cr in the air.

- Line 114: Clarify recovery adjustments. Clarify whether recovery-adjusted data are reported . If yes, state correction methodology; if no, justify.

Authors' response:

Thank you for your comment. The use of recovery information in analytical measurements is often optional. In our case, we were guided by the suitability of the measurement data for the specific purpose. We decided to use the original data in the manuscript because the recovery is high enough (Cr - 95.2% and Cr(VI) - 99.3%) and our results are not related to the enforcement analysis (the difference between applying or not applying a correction factor to the measurement data does not mean that a legal limit is exceeded or that a result is in compliance with the limit).

**Results & Discussion**

- Line 155-160 Include temperature, humidity, precipitation, and wind speed statistics to contextualize seasonal trends.

Authors' response:

Thank you for your comment. Averaged weather conditions for each weekly sampling cycle are presented in Table S2 (Supplementary Material), which has been added to the article. All changes resulting from the addition of the table have been made throughout the article.

- Line 174-180 The discussion should better situate Radom's Cr levels within broader European urban air pollution trends. How do the observed concentrations compare to other Polish or EU cities with similar industrial/traffic profiles?

And

- Line 184-189 Provide more detailed hypotheses for seasonal trends (e.g., winter increases due to heating emissions, summer decreases due to atmospheric dispersion). Link these to meteorological data (e.g., inversion events, wind patterns).

Authors' response:

Thank you for your comments. Both paragraphs have been edited. Current version (lines 174-183 and 187-197): Cr$_{tot}$ concentrations in the particulate matter fractions studied ranged from 0.08 to 4.09 ng/m$^3$. Like in the case of particulate matter concentrations, the results were several times lower than the chromium concentrations given in our previous work for the non-industrial zone (15 ng/m$^3$ on average), when we studied TSP (Świetlik et al., 2011). It should be pointed out that particulate matter pollution has been substantially reduced in Poland in recent years, owing to the application of state-of-the-art, efficient, and environment-friendly technological solutions. The continuing modernization of the energy, heating, and industrial sectors - such as the EU Clean Air Program (since 2018) and provincial anti-smog resolutions (since 2017) - along with improved fuel quality regulations (established by the Minister of Industry and the Minister of Climate and Environment regarding the quality of solid fuels since 2018), has led to a consistent reduction in the amount of particulate matter pollution emitted into the air each year

(EU Clean Air Program, 2024; Regulation of the Minister of Industry and the Minister of Climate and Environment, 2024).

The maximum $Cr_{tot}$ concentration relating to $PM_{10}$ was found in winter ($2.23\pm0.53$ ng/m$^3$ on average), whereas it averaged $1.71\pm0.83$ ng/m$^3$ in the entire measurement period. The mean concentrations relating to $PM_{10}$ in Radom were similar to those determined in other cities in Europe in urban areas: Edinburgh (U.K.) 1.6 ng/m$^3$ (Heal et al., 2005), Katowice (Poland) 2.39 ng/m$^3$ (Rogula-Kozłowska, 2015), Rome (Italy) 2-5 ng/m$^3$ (Catrambone et al., 2013).

The $PM_{2.5}$ fraction contains approximately 80% of the total chromium content. A similar correlation was observed for the PM concentration in the winter season, where 86% of $PM_{10}$ was the $PM_{2.5}$ fraction (Table 1). The $PM_{2.5}$ fraction is assumed to be the result of emissions from anthropogenic sources (Nocoń et al., 2018). The sampling point was located near a busy street and a single-family housing estate (from the west and south-west) and far away from an industrial zone (approx. 8-10 km from the south and south-west). Considering that Radom is a city with prevailing westerly winds, especially in autumn and winter (WeatherSpark, 2025), it can be supposed that municipal emissions, mainly stationary coal combustion sources, and road traffic are probably the main factors influencing the measured chromium concentrations.

- Section 3.4 The risk assessment is a strength, but it should explicitly compare calculated hazard quotients (HQs) or cancer risks with regulatory thresholds (e.g., WHO, EU limits). Discuss how findings align with EU air quality directives (e.g., compliance with Cr(VI) thresholds).

Authors' response:

Thanks for your comment. The discussion of results in section 3.4 has been revised. Current version (lines 288-314):

The estimated potential non-carcinogenic (HQ) and carcinogenic (CR) risk of inhalatory exposure to Cr and Cr(VI) present in $PM_{10}$ for urban residents is shown in Table 3. Both were maximum in winter, when Cr concentrations become highest. The World Health Organization (WHO) recommends the baseline limit of hexavalent chromium with an excess lifetime risk (RR values) of 1:10,000, 1:100,000, and 1:1,000,000 to be 2.5, 0.25, and 0.025 ng/m$^3$, respectively (WHO, 2000). In the light of the standard interpretation, however, regardless of the season, the carcinogenic risk to residents based on Cr(VI) concentration in $PM_{10}$ was below the US EPA threshold for significant risk (between $1\cdot10^{-6}$ and $1\cdot10^{-4}$) and amounted to between $1.11\cdot10^{-6}$ and $5.78\cdot10^{-6}$ for children and from $3.69\cdot10^{-6}$ to $1.92\cdot10^{-5}$ for adults. Even in the case of maximum Cr(VI) concentration in $PM_{10}$ during the winter season, the estimated carcinogenic risk to the population of Radom was lower than the upper limit: CR=$9.24\cdot10^{-6}$ for children and CR=$3.12\cdot10^{-5}$ for adults. However, although the risk of cancer in children is generally low, it can be misleading because exposures during development can have disproportionately dramatic effects in adulthood.

The estimated non-carcinogenic inhalation risk from chromium for the residents of Radom posed little risk, either. The HQ values calculated based on the total Cr concentration in $PM_{10}$ were lower than the safe level (HQ = 1) and ranged from $1.32\cdot10^{-2}$ to $2.14\cdot10^{-2}$, indicating little non-carcinogenic risk from chromium. The non-carcinogenic health risk based on the maximum $Cr_{tot}$ concentration in $PM_{10}$ was low, too, as it was not more than one (Table 3). These risk values are no different from those found in the literature, e.g., HQ in Italy was in the range $1.5\cdot10^{-2}$ – $5.4\cdot10^{-2}$ (Diana et al., 2023).

EU air quality directives do not provide for separate normative values for chromium. Directive 2024/2881 specifies quality objectives for arsenic, cadmium, lead, mercury, nickel, and PAHs, but does not cover chromium (Directive 2024/2881). Within the framework of EU policy, chromium(VI) is regulated mainly in the context of occupational safety – the binding occupational exposure limit has been set at 0.005 mg/m$^3$ by 2025 (Directive 2017/2398). For comparison, the highest obtained concentration of Cr(VI) in $PM_{10}$ (1.3544 ng/m$^3$) is almost three orders of magnitude lower than this value. Consequently, the estimated health risk of chromium is low and does not raise significant health concerns.

In conclusion, the estimated potential non-carcinogenic (HQ) and carcinogenic (CR) risk of inhalatory exposure to Cr and Cr(VI), respectively, present in $PM_{10}$ for the urban residents of Radom indicates that risks of adverse health effects from chromium and chromium(VI) exposure in Radom are relatively low and complies with the applicable WHO and EU regulatory framework (WHO, 2021; Directive 2017/2398).

*With gratitude*

*Monika Łożyńska*

---

## Author Response (AR2)

**Response on RC1 (Round 2)**

about manuscript:

"Measurement Report: Seasonal trends and chemical speciation of chromium(III/VI) in different fractions of urban particulate matter – a case study of Radom, Poland" Monika Łożyńska, Marzena Trojanowska, Artur Molik, Ryszard Świetlik.

Thank you very much for the suggestions and comments contained in the review. All suggested changes have been included in the manuscript. Answers to the comments are presented below:

**Specific Comments:**

**Abstract**

- Line 18: Change "varied a lot…" to "varied from a low in summer (…) to a high in winter (…).". Consider adding a sentence to the end of the abstract highlighting the importance of this measurement report.

Authors' response:

As suggested by the reviewer, the sentences have been corrected. Current version (lines 16-17): The Cr(VI) share in PM in the particular seasons varied from a low in summer (9.1% of Crtot) to a high in winter (40% of Crtot). A concluding sentence has also been added. Current version (lines 21-22): The studies presented in manuscript fill the research gap of chromium and hexavalent chromium measurements in particulate matter of different sizes in the air of a medium-sized city in central Poland.

**Introduction**

- Line 36: Add the word "mostly" after the parenthetical to say PM2.5 "mostly comes from anthropogenic processes".

Authors' response:

As suggested by the reviewer, the sentences have been corrected. Current version (lines 30-32): The particulate matter's coarse fraction (2.5-10 μm) is assumed to be of natural origin, while the fine fraction (0.1–2.5 μm, especially the particles of less than 1 μm) mostly comes from anthropogenic processes (Nocoń et al., 2018).

- Lines 65 – 66: Delete the first sentence of this paragraph, as the second sentence regarding average atmospheric concentrations of Cr is more specific and relevant.

Authors' response:

At the reviewer's suggestion, the sentence has been removed.

- Line 71: After "Cr(VI)" add the words "concentrations in the United States" for clarity.

Authors' response:

As suggested by the reviewer, the sentences have been corrected. Current version (lines 62-63): According the US EPA National Air Toxics Assessment in 2017, the median, mean, and maximum Cr(VI) concentrations in the United States were 0.03 ng/m3, 0.1 ng/m3 and 3.18 ng/m3, respectively (Proctor et al., 2021).

- Line 78: Change "unavailable" to "limited".

Authors' response:

As suggested by the reviewer, the sentences have been corrected. Current version (lines 67-69): Although chromium occurrence in urban air has been extensively studied and a range of publications have appeared recently (Catrambone et al., 2013; Widziewicz et al., 2016; Nocoń et al., 2018), investigations of total Cr occurrence and its valence speciation in particulate matter of different particle sizes are still limited.

**Experimental**

- Line 93: Consider adding tanneries to this list, as mentioned at the end of the introduction.

Authors' response:

Thank you for your comment. Tanneries have been added to the list. Current version (lines 82-83): The local sources of chromium emissions are: road traffic, coal burning in homes, coal-fired municipal heating plants, tanneries and multiple metalworks.

- Line 100: It is very helpful to see the weather data separated by sampling week. However, precipitation is still missing from Table S2. This is important to include as it informs the reader how much chromium might have been lost due to wet deposition over that sampling period.

Authors' response:

Following the reviewer's suggestion and the comment in 'General Comments', Table S2 has been supplemented.

- Line 104: Per the author's "Reply on RC2", this air rate should be changed to state "The sampling rate was maintained at approximately 6 m3/h."

Authors' response:

Thanks for your comment. The flow rate data has been changed. Current version (line 94): The sampling rate was maintained at approximately 6 $m^3$/h.

- Line 105: Change the word "weight" to "precision" for clarity of the instrument used.

Authors' response:

As suggested by the reviewer, the sentences have been corrected. Current version (lines 95-96): The filters were weighed before and after sampling, the precision was 0.01 mg (Microbalance MX5 Mettler Toledo) in a temperature and relative humidity controlled environment (20±3°C and 50±10%, respectively).

- Line 145: Per the author's "Reply on RC2", revise or add an additional sentence that since "the recovery is high enough (Cr - 95.2% and Cr(VI) - 99.3%) and our results are not related to" enforcement," the data presented in this report are uncorrected for sample recovery efficiency. If only presented here, it should be clear that the claim refers to both Crtot and Cr(VI).

Authors' response:

At the reviewer's suggestion, sentence has been added. Current version (lines 134-136): The recovery is high enough (Cr - 95.2% and Cr(VI) - 99.3%) and our results are not related to the enforcement analysis, which is why the data presented in this report are uncorrected for sample recovery efficiency.

**Results**

- Line 187 – 194: This new section is very helpful, but would be better suited to be added as a final paragraph to the preceding section (3.1) since it refers to all PM, not just Cr. The authors could add a sentence in this paragraph in Section 3.2 that suggests Cr levels in PM are decreasing due to the same reasons that PM is decreasing, as well as the role of their current sampling location. This is already

mentioned in lines 208 – 210, though, so an additional statement in Lines 187 – 194 may be unnecessary. (Comment from the "Technical Corrections")

Authors' response:

Thanks for your comment. The paragraph "It should be..." has been moved to section 3.1. In section 3.2 (lines 171-177), an appropriate explanatory sentence has been added (lines 184-186): In Poland, the concentrations of Cr in the PM, similar to the particulate matter concentrations, decrease every year thanks to the use of state-of-the-art, efficient and environmentally friendly technological solutions.

- Lines 304 – 305: This change mentions "is likely to cause carcinogenic effects in the future from lifetime exposure" which should be changed to "poses a substantial risk of carcinogenic effects from lifetime exposure".

Authors' response:

As suggested by the reviewer, the sentences have been corrected. Current version (lines 286-289): A carcinogenic risk value above the upper limit ($1 \cdot 10^{-4}$) suggests that chromium(VI) in atmospheric particulate matter poses a substantial risk of carcinogenic effects in the future from lifetime exposure, while values below the lower limit ($1 \cdot 10^{-6}$) do not pose a significant risk. HQ below one suggests no significant risk of non-carcinogenic effects.

- Lines 311 – 313: The sentence "The World Health Organization…" should be deleted here as it is a repeat of the sentence on Lines 258 – 260.

Authors' response:

At the reviewer's suggestion, the sentence has been removed.

**Technical corrections:**

Thank you so much for all your suggestions. The authors agree with all the reviewer's comments in the "Technical Corrections" section. All suggested changes are incorporated into the manuscript.

*With gratitude*

*Monika Łożyńska*